# THINK TWICE BEFORE YOU ACT: IMPROVING INVERSE PROBLEM SOLVING WITH MCMC

## ABSTRACT

Recent studies demonstrate that diffusion models can serve as a strong prior for solving inverse problems. A prominent example is Diffusion Posterior Sampling (DPS), which approximates the posterior distribution of data given the measure using Tweedie's formula. Despite the merits of being versatile in solving various inverse problems without re-training, the performance of DPS is hindered by the fact that this posterior approximation can be inaccurate especially for high noise levels. Therefore, we propose **D**iffusion **P**osterior **MCMC** (**DPMC**), a novel inference algorithm based on Annealed MCMC to solve inverse problems with pretrained diffusion models. We define a series of intermediate distributions inspired by the approximated conditional distributions used by DPS. Through annealed MCMC sampling, we encourage the samples to follow each intermediate distribution more closely before moving to the next distribution at a lower noise level, and therefore reduce the accumulated error along the path. We test our algorithm in various inverse problems, including super resolution, Gaussian deblurring, motion deblurring, inpainting, and phase retrieval. Our algorithm outperforms DPS with less number of evaluations across nearly all tasks, and is competitive among existing approaches.

## 1 INTRODUCTION

Diffusion Models (Sohl-Dickstein et al., 2015; Ho et al., 2020; Song & Ermon, 2019; Song et al., 2021b) have recently achieved significant success in high-dimensional data generation, including images (Dhariwal & Nichol, 2021; Ramesh et al., 2022; Rombach et al., 2022; Saharia et al., 2022), videos (Ho et al., 2022b; OpenAI, 2024; Ho et al., 2022a; Blattmann et al., 2023; Girdhar et al., 2023), audio Kong et al. (2020); Chen et al. (2020), text Li et al. (2022), and and 3D generation (Liu et al., 2023; Poole et al., 2023; Wu et al., 2023; Shi et al., 2023; Gao et al., 2024). Beyond generation (Zhang et al., 2024a; Chang et al., 2024), recent works have applied diffusion models to solve inverse problems in a plug-and-play fashion without the need for fine-tuning (Jalal et al., 2021; Kadkhodaie & Simoncelli, 2021; Kawar et al., 2022; Song et al., 2021b; Choi et al., 2021; Chung et al., 2022a; 2023a; Dou & Song, 2023; Mardani et al., 2023a; Song et al., 2023b; Rout et al., 2023; Song et al., 2023a; Chung & Ye, 2022; Feng et al., 2023; Zhu et al., 2023b; Chung et al., 2023b; Zhang et al., 2024b), where the goal is to restore data $\mathbf{x}$ from degraded measurements $\mathbf{y}$. Among these, one line of works Jalal et al. (2021); Chung et al. (2023a); Song et al. (2023b); Rout et al. (2023); Song et al. (2023a) proposes to modify the inference process of diffusion models with guidance that encourages samples to be consistent with the measure. Other perspectives include variational inference (Mardani et al., 2023a; Feng et al., 2023; Zhu et al., 2023b), Bayesian filtering Dou & Song (2023), and solving an inner loop optimization problem during sampling (Song et al., 2022; 2023a).

A typical challenge in this context is that the posterior distribution $p(\mathbf{x}|\mathbf{y})$ is only defined for clean samples $\mathbf{x}$, yet during sampling, an estimation of $p(\mathbf{x}_t|\mathbf{y})$ is needed at each diffusion step $t$. DPS (Chung et al., 2023a) approximates the intractable posterior through Tweedie's formula (Efron, 2011), enabling its application in general inverse problems. However, this approximation might be inaccurate especially on high noise levels, leading to samples of low quality. To address this, $\Pi$GDM (Song et al., 2023b) attempts to improve the approximated guidance by pseudo-inverting the measurement. Additionally, Rout et al. (2023); Song et al. (2023a) aim to enhance DPS in the Latent Diffusion model (Rombach et al., 2022) by introducing extra guidance terms or resampling processes to the data space.

In this work, we propose to leverage annealed Markov Chain Monte Carlo (MCMC) to improve solving inverse problems with diffusion models. As a family of effective algorithms that draw samples from complex distributions, MCMC is widely used in training Energy-Based Models (EBMs) (Xie et al., 2016; Nijkamp et al., 2019; Du et al., 2021) and sampling from diffusion models (Song & Ermon, 2019; Song et al., 2021b). Annealed MCMC further proposes to facilitate MCMC sampling by gradually sampling from a sequence of distributions from decreasingly lower temperature, to accelerate the mixing. Previous work (Du et al., 2023) has leveraged Annealed MCMC in compositional generation with EBMs or diffusion models. In this work, we propose using MCMC to reduce the error of posterior approximation in solving inverse problems.

There are some earlier works (Jalal et al., 2021; Song et al., 2021b) that employ annealed MCMC with Langevin dynamics to solve inverse problems. They tackle the intractability of the posterior distribution by projecting the current sample onto the measurement subspace. However, these approaches might fail when measurements are noisy or the measurement process is non-linear, as discussed in Chung et al. (2023a). Moreover, a single MCMC step is executed at every noise level before moving to the next noise level. Alternatively, we propose to build the the intermediate distributions of our annealed MCMC with the approximated posterior distributions derived in Chung et al. (2023a). Despite the fact that such approximation might be inaccurate for the data distributions defined by the forward diffusion, they still remain as a valid sequence of distributions that can be leveraged in annealed MCMC. We further propose to run multiple sampling steps at each noise level similar to Du et al. (2023), that empirically improves the performance.

In summary, we make the following contributions in this work:

- We propose the **D**iffusion **P**osterior **MCMC** (**DPMC**) algorithm, which leverages annealed MCMC with a sequence of posterior distribution of data given measurements, whose formula is inspired by DPS (Chung et al., 2023a).

- We demonstrate that empirically DPMC outperforms DPS in terms of sample quality across various types of inverse problems in image domain, including both linear and nonlinear inverse problems. DPMC also establishes competitive performance compared with other existing approaches.

- Through extensive ablation study and comparison with other approaches, we highlight the effectiveness of MCMC-based approaches for solving inverse problems, and a constant improvement of performance with increasingly larger number of sampling step.

## 2 BACKGROUND

### 2.1 INVERSE PROBLEM

We denote data distribution $\mathbf{x} \sim p_{data}(\mathbf{x})$. In many scientific applications, instead of directly observing $\mathbf{x}$, we might only have a partial measurement $\mathbf{y}$, which is derived from $\mathbf{x}$, and we want to restore $\mathbf{x}$ from $\mathbf{y}$. Formally, we might assume the following mapping between $\mathbf{x}$ and $\mathbf{y}$

$$\mathbf{y} = \mathbf{A}(\mathbf{x}) + \mathbf{n}, \ \mathbf{x} \in \mathbb{R}^D, \ \mathbf{y} \in \mathbb{R}^d, \ \mathbf{n} \sim \mathcal{N}(0, \sigma^2 \mathbf{I}) \tag{1}$$

where $\mathbf{A}(\cdot) : \mathbb{R}^D \mapsto \mathbb{R}^d$ is called forward measurement operator and $\mathbf{n}$ is the measurement noise following Gaussian distribution. Therefore, we have $p(\mathbf{y}|\mathbf{x}) \sim \mathcal{N}(\mathbf{A}(\mathbf{x}), \sigma^2 \mathbf{I})$. Mapping between $\mathbf{x}$ to $\mathbf{y}$ can be many-to-one. This makes exactly restoring $\mathbf{x}$ become an ill-posed problem.

### 2.2 DIFFUSION MODELS

Diffusion models (Sohl-Dickstein et al., 2015; Ho et al., 2020; Song et al., 2022; Song & Ermon, 2019) define a generative process that gradually transforms a random noise distribution into a clean data distribution. A diffusion model consists of a forward noise injection process and a backward denoising process. Let $\mathbf{x}_0 \sim p_{data}$ denote clean observed samples. DDPM (Ho et al., 2020) defines a Markovian forward process as follows:

$$q(\mathbf{x}_{1:T}|\mathbf{x}_{t-1}) = q(\mathbf{x}_t|\mathbf{x}_{t-1}) = \mathcal{N}(\sqrt{\alpha_t}\mathbf{x}_{t-1}, \beta_t \mathbf{I})$$
$$q(\mathbf{x}_t|\mathbf{x}_0) = \mathcal{N}(\sqrt{\bar{\alpha}_t}\mathbf{x}_0, (1 - \bar{\alpha}_t)\mathbf{I}) \tag{2}$$

where $\{\beta_t\}_{t=1}^T$ is the manually designed noise schedule that might differ from different works (Ho et al., 2020; Song et al., 2021b; Karras et al., 2022). And $\alpha_t = 1 - \beta_t$, $\bar{\alpha}_t = \prod_{i=1}^t \alpha_i$. In the backward process, we start from the noise distribution and gradually denoise the samples as follows:

$$\mathbf{x}_{t-1} = \mu_\theta(\mathbf{x}_t, t) + \sqrt{\tilde{\beta}_t}\mathbf{z}_t, \ \ \mathbf{z}_t \sim \mathcal{N}(0, \mathbf{I})$$

$$\mu_\theta(\mathbf{x}_t, t) = \frac{1}{\sqrt{1-\beta_t}}\left(\mathbf{x}_t - \frac{\beta_t}{\sqrt{1-\bar{\alpha}_t}}\epsilon_\theta(\mathbf{x}_t, t)\right), \ \ \tilde{\beta}_t = \frac{1-\bar{\alpha}_{t-1}}{1-\bar{\alpha}_t}\beta_t \tag{3}$$

where $\epsilon_\theta(\mathbf{x}_t, t)$ is parameterized by a neural network (Ho et al., 2020; Peebles & Xie, 2023). Let the marginal distribution defined by the forward process of $\mathbf{x}_t$ be denoted as $p_t(\mathbf{x}_t)$. When trained with denoising score matching loss (Ho et al., 2020; Song et al., 2021b), with sufficient data and model capacity, for almost all $\mathbf{x}$ and $t$, $\epsilon_\theta(\mathbf{x}_t, t)$ corresponds to the gradient field $\nabla_{\mathbf{x}_t}\log p_t(\mathbf{x}_t)$ as follows:

$$\nabla_{\mathbf{x}_t}\log p_t(\mathbf{x}_t) = -\frac{\epsilon_\theta(\mathbf{x}_t, t)}{\sqrt{1-\bar{\alpha}_t}} \tag{4}$$

Other than DDPM, DDIM (Song et al., 2021a) defines a Non-Markovian forward process that shares the same training objective as DDPM. Thus, a model trained with DDPM can be directly applied in the DDIM sampler to accelerate the sampling process. A DDIM sampler follows:

$$\mathbf{x}_{t-1} = \sqrt{\bar{\alpha}_{t-1}}\left(\frac{\mathbf{x}_t - \sqrt{1-\bar{\alpha}_t}\epsilon_\theta(\mathbf{x}_t, t)}{\sqrt{\bar{\alpha}_t}}\right) + \sqrt{1 - \bar{\alpha}_{t-1} - \sigma_t^2}\epsilon_\theta(\mathbf{x}_t, t) + \sigma_t\mathbf{z}_t \tag{5}$$

where $\mathbf{z}_t \sim \mathcal{N}(0, \mathbf{I})$ and variance $\sigma_t$ can be arbitrary defined.

### 2.3 DIFFUSION POSTERIOR SAMPLING

To solve the ill-posed inverse problem, Diffusion Posterior Sampling (DPS) (Chung et al., 2023a) recruits pretrained diffusion models (Ho et al., 2020; Song et al., 2021b) as prior and propose an iterative optimization algorithm. According to Bayes rule, the gradient field of posterior distribution $\nabla_{\mathbf{x}_t}\log p(\mathbf{x}_t|y) = \nabla_{\mathbf{x}_t}\log p_t(\mathbf{x}_t) + \nabla_{\mathbf{x}_t}\log p(\mathbf{y}|\mathbf{x}_t)$ where the term $\nabla_{\mathbf{x}_t}\log p_t(\mathbf{x}_t)$ is estimated by the pretrained diffusion model using equation 4. In DPS, the authors adapt the approximation $p(\mathbf{y}|\mathbf{x}_t) \approx p(\mathbf{y}|\hat{\mathbf{x}}_0)$, where $\hat{\mathbf{x}}_0(\mathbf{x}_t) = E_{\mathbf{x}_0 \sim p(\mathbf{x}_0|\mathbf{x}_t)}[\mathbf{x}_0]$ can be estimated through Tweedie's formula (Efron, 2011):

$$\hat{\mathbf{x}}_0 = \frac{\mathbf{x}_t - \sqrt{1-\bar{\alpha}_t}\epsilon_\theta(\mathbf{x}_t, t)}{\sqrt{\bar{\alpha}_t}} \tag{6}$$

Given noisy observation data $x_t$, DPS makes the following updates

$$\mathbf{x}'_{t-1} = \frac{1}{\sqrt{1-\beta_t}}(\mathbf{x}_t - \frac{\beta_t}{\sqrt{1-\bar{\alpha}_t}}\epsilon_\theta(\mathbf{x}_t, t)) + \sigma_t\mathbf{z}_t, \ \ \mathbf{z}_t \sim \mathcal{N}(0, \mathbf{I})$$

$$\mathbf{x}_{t-1} = \mathbf{x}'_{t-1} - \zeta_t\nabla_{\mathbf{x}_t}\|\mathbf{y} - \mathbf{A}(\hat{\mathbf{x}}_0)\|_2^2 \tag{7}$$

In practice, DPS employs the following adaptive step size parameter $\zeta_t$:

$$\zeta_t = \frac{\zeta}{\|\mathbf{y} - \mathbf{A}(\hat{\mathbf{x}}_0)\|_2} \tag{8}$$

where $\zeta$ is a fix constant.

## 3 DIFFUSION POSTERIOR WITH MCMC SAMPLING

In this section, we introduce our DPMC model, which combines diffusion models and Markov Chain Monte Carlo (MCMC) sampling. The former one is for the progressive denoising to provide an unconditional proposal distribution. The latter is for the conditional guidance by the measurement $\mathbf{y}$. According to *Theorem 1* in Chung et al. (2023a), error of the approximation $p(\mathbf{y}|\mathbf{x}_t) \approx p(\mathbf{y}|\hat{\mathbf{x}}_0)$ is bounded by the estimation error between $\mathbf{x}_0$ and $\hat{\mathbf{x}}_0$. When the noise level is low, $p(\mathbf{x}_0|\mathbf{x}_t)$ can be single-modal and the estimation $\hat{\mathbf{x}}_0$ and the posterior approximation might be accurate. However,

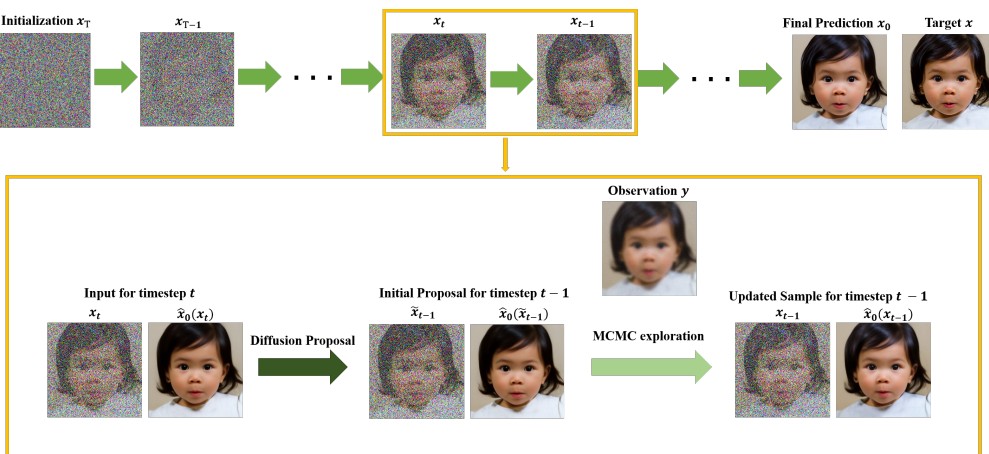

Figure 1: The illustration of DPMC. At each step, DPMC iterates over diffusion proposal step and MCMC exploration step.

at higher noise levels, where $p(\mathbf{x}_0|\mathbf{x}_t)$ can be indeed multi-modal and $\hat{\mathbf{x}}_0$ can be far from $\mathbf{x}_0$, this approximation can be too loose, potentially leading to inferior results. As illustrated in Figure 2, while the samples of DPS are valid, they might fail to capture vivid details. To address these problems, we instead resort to MCMC sampling.

Unlike diffusion models, which explicitly define all the noise distributions $p_t(\mathbf{x}_t)$ through the forward process, MCMC sampling does not require knowledge of the sample distributions at each intermediate MCMC step. As long as a valid transition kernel and a sufficient number of MCMC sampling steps are used, MCMC is known to sample from arbitrary underlying distributions.

Annealed MCMC (Neal, 2001; Song & Ermon, 2019; Song et al., 2021b) is a widely used technique to accelerate the MCMC sampling process for highly multi-modal data. In the annealed sampling process, samples gradually progress through a series of intermediate distributions with different temperatures. MCMC is applied at each intermediate distribution to enable samples to transition from the previous distribution to the current one. In this work, we apply annealed MCMC to solve the inverse problem. We introduce a series of intermediate distributions $\{\tilde{p}_t(\mathbf{x}_t|\mathbf{y})\}_{t=1}^T$. Note that we do not expect $\tilde{p}_t(\mathbf{x}_t|\mathbf{y})$ to be close to the true posterior distribution $p_t(\mathbf{x}_t|\mathbf{y})$ at every intermediate distribution. Instead, we only require $\tilde{p}_t(\mathbf{x}_t|\mathbf{y})$ to agree with $p_t(\mathbf{x}_t|\mathbf{y})$ at

---

**Algorithm 1** DPMC Algorithm

**Require:** Inverse problem forward operator $\mathcal{A}(.)$, noisy measurement $\mathbf{y}$, pretrained diffusion prior model $p_t(\mathbf{x}_t)$, number of intermediate noise levels $T$, number of MCMC steps $K$, MCMC step size $\eta_t$, weighting parameter $\xi_t$.
1: $\mathbf{x}_T \sim \mathcal{N}(0, \mathbf{I})$
2: **for** $t = T$ to 1 **do**
3:     **Proposal step:** Sample $\tilde{\mathbf{x}}_{t-1}$ from $\mathbf{x}_t$ following Eq. 5
4:     **Exploration step:** Set $\mathbf{x}_{t-1}^{(0)} = \tilde{\mathbf{x}}_{t-1}$
5:     **for** $k = 1$ to $K$ **do**
6:         Update $\mathbf{x}_{t-1}^k$ to $\mathbf{x}_{t-1}^{k+1}$ following Eq. 10
7:     **end for**
8: **end for**

---

the clean image distribution. The key idea is that MCMC sampling can bridge the gap between different intermediate distributions. And the intermediate distributions only need to form a trajectory that enables MCMC to smoothly transition from the noise distribution to the target distribution in the clean data space.

Following Chung et al. (2023a), we define the intermediate distributions $\tilde{p}_t(\mathbf{x}_t|y)$ as

$$\tilde{p}_t(\mathbf{x}_t|\mathbf{y}) \propto p_t(\mathbf{x}_t) \exp(-\rho\|\mathbf{y} - \mathbf{A}(\hat{\mathbf{x}}_0)\|_2^2) \tag{9}$$

where $\rho = 1/\sigma^2$. $p_t(\mathbf{x}_t)$ denotes the diffusion prior at noise level $t$ and $\nabla_{\mathbf{x}_t} \log p_t(\mathbf{x}_t)$ can be estimated using equation 4. However, instead of using 7 for sampling, which requires the intermediate distribution to be sufficiently close to the ground truth posterior at each noise level, we propose a new proposal-and-update algorithm based on annealed MCMC sampling.

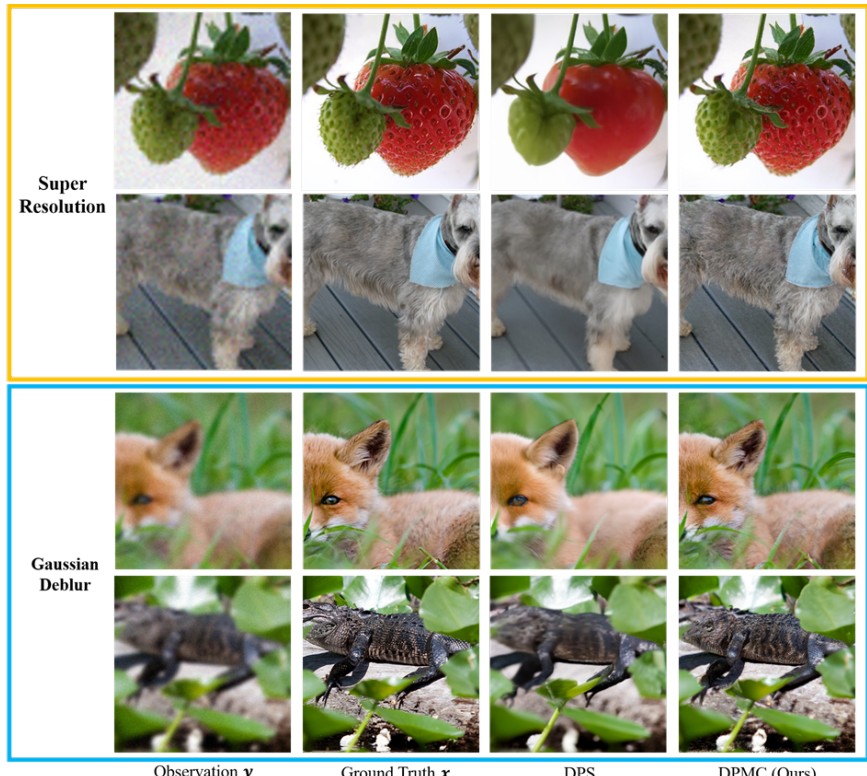

Figure 2: Qualitative comparison between DPS samples and our DPMC samples. Images in the top rows are from super resolution task. Images in the bottom rows are from Gaussian deblurring task.

**Proposal Stage:** Given samples $\mathbf{x}_t$ that follow the intermediate distribution $\tilde{p}_t(\mathbf{x}_t|\mathbf{y})$, we first denoise them to $t-1$ following the standard diffusion step without considering the extra guidance. We denote the proposed sample as $\tilde{\mathbf{x}}_{t-1}$. The proposal step aims to provide a good initialization for the intermediate distribution at $\tilde{p}_{t-1}(\mathbf{x}_{t-1}|\mathbf{y})$ with the help of the diffusion model. The samples $\tilde{\mathbf{x}}_{t-1}$ might not fully adhere to the target distribution $\tilde{p}_{t-1}(\mathbf{x}_{t-1}|\mathbf{y})$, but we hope they are close enough to the target distribution so that we can obtain good samples with a few MCMC sampling steps.

**Exploration Stage:** We then encourage the samples to explore the landscape at noise level $t-1$ and follow $\tilde{p}_{t-1}(\mathbf{x}_{t-1}|\mathbf{y})$ with MCMC updates. We employ Langevin Dynamics (Roberts & Tweedie, 1996) as the transition kernel. Specifically, starting from $\mathbf{x}_{t-1}^0 = \tilde{\mathbf{x}}_{t-1}$, we iterate the following updates:

$$\mathbf{x}_{t-1}^{k+1} = \mathbf{x}_{t-1}^k + \eta_{t-1}\nabla_{\mathbf{x}_{t-1}^k}\log\tilde{p}_{t-1}(\mathbf{x}_{t-1}^k|\mathbf{y}) + \sqrt{2\eta_{t-1}}\mathbf{w}, \ \mathbf{w}\sim\mathcal{N}(0,\mathbf{I}) \qquad (10)$$

We named our algorithm **D**iffusion **P**osterior **MC**MC (**DPMC**). We illustrate DPMC Figure 1 and in Algorithm 1. We also provide a theoretical analysis which shows that the conditional distribution derived by DPMC is $\varepsilon$-close to the ground truth, i.e.

$$D_{\text{TV}}(q_0(\mathbf{x}_0 \mid \mathbf{y}) \parallel p^*(\mathbf{x}_0 \mid \mathbf{y})) \leq \varepsilon$$

with a carefully chosen step size $\eta_t$ and inner loop $K$ of the Langevin MCMC algorithm, under several assumptions on the score estimation error, conditional probability approximation error as well as some convexity and Lipschitz continuity conditions. Detailed assumptions and main theorem for the convergence of our DPMC algorithm is deferred to Appendix B.

**Implementation Details** We apply the same implementation as Chung et al. (2023a) in equation 8 by introducing a step size parameter $\zeta_t$ to be inverse proportional to optimization distance $\|\mathbf{y} - \mathbf{A}(\hat{\mathbf{x}}_0)\|_2$. This setting equals to sampling according to the prior distribution tilted by an exponential distribution of the optimization distance. In practice, we calculate $\nabla_{\mathbf{x}_t}\tilde{p}_t(\mathbf{x}_t|\mathbf{y})$ at each step using the following equation:

$$\nabla_{\mathbf{x}_t} \log \tilde{p}_t(\mathbf{x}_t|\mathbf{y}) = \nabla_{\mathbf{x}_t} \log p_t(\mathbf{x}_t) - \xi_t \nabla_{\mathbf{x}_t} \|\mathbf{y} - \mathbf{A}(\hat{\mathbf{x}}_0)\|_2 \tag{11}$$

where $\xi_t$ represents a constant or variable weighting schedule. Note that we replace square $l_2$ norm by $l_2$ norm itself. According to Chung et al. (2023a); Song & Dhariwal (2023), using $l_2$ norm might make the optimization to be more robust to outliers and can achieve more stable results than square $l_2$ norm as it imposes a smaller penalty for large errors.

While adding MCMC exploration might inevitably introduce more sampling steps at each intermediate distribution and thus increase the sampling time, we find that with DPMC, we can counteract this by reducing the number of intermediate distributions needed. In fact, we find that using 200 intermediate distributions is sufficient for DPMC. Moreover, during the early stage of diffusion, the samples are very close to Gaussian noise. Considering that the reason for inserting intermediate distributions is to provide a good initialization for sampling more complex distributions that are very different than Gaussian, we can skip those early stages with high noise levels and start MCMC sampling directly from a moderate noise level. On the other hand, at very low noise levels, $p(\mathbf{x}_0|\mathbf{x}_t)$ almost becomes single-modal, and the approximation $p(\mathbf{y}|\mathbf{x}_t) \sim p(\mathbf{y}|\mathbf{x}_0)$ becomes sufficiently accurate. Thus, we can directly follow DPS without the need for additional MCMC sampling. In fact, we find that DPMC performs well when we apply the proposal and exploration steps only to the middle steps of the sampling process, while following the original DPS setting at both ends. This further reduces the number of evaluation steps needed. In practice, DPMC can achieve much better results compared to DPS with an even smaller number of evaluations (NFEs).

## 4 EXPERIMENTS

In this section, we conduct experiments on the DPMC method proposed by us, which outperforms existing baselines on diffusion posterior sampling. To start with, we thoroughly introduce the models, datasets and settings of our experiments.

### 4.1 EXPERIMENT SETTINGS

**Datasets and Pretrained Model:** Following Chung et al. (2023a), we test our algorithm on FFHQ $256 \times 256$ dataset (Karras et al., 2019) and ImageNet $256 \times 256$ dataset (Deng et al., 2009). Same as Chung et al. (2023a); Dou & Song (2023), we use 1k validation images for each dataset. For FFHQ dataset, we use the pretrained model provided by Chung et al. (2023a). For ImageNet dataset, we use the pretrained model provided by Dhariwal & Nichol (2021).

**Inverse Problems:** We evaluate the effectiveness of our DPMC algorithm on the following inverse problems: (i) Super-resolution with 4x bicubic downsampling as the forward measurement; (ii) Random inpainting using both box masks and random masks; (iii) Deblurring with Gaussian blur kernels and motion blur kernels [1]; (iv) Phase retrieval, where we perform a Fourier transformation on each image and use only the Fourier magnitude as the measurement. Among these inverse problems, (i), (ii), and (iii) are linear inverse problems, while (iv) is a nonlinear inverse problem. We use Gaussian noise with $\sigma = 0.05$ for all tasks and set task-related parameters according to Chung et al. (2023a); Dou & Song (2023) to ensure a fair comparison.

**Hyper-Parameter Setting:** We use the DDIM sampler with default variance $\sigma_t = 0$ as our diffusion proposal sampler. We use $T = 200$ intermediate distributions and $K = 4$ MCMC steps at each intermediate distribution. Empirically, we set $\eta_t = \eta\beta_t$ and $\xi_t = \xi\bar{\alpha}_t^3$, where $\eta$ and $\xi$ are task-related constants. We find these schedules work well across all settings. The detailed parameter settings can be found in Section A in the Appendix. As discussed in 3, we apply the proposal-exploration step in the middle 60% of the total sampling steps and resort to original DPS sampler step at the first 30% and the last 10% noise levels. An exception is the inpainting task with a box-type mask on the ImageNet dataset, where we find that applying the proposal-exploration step until clean images yields better results. Considering both the initial proposal step and the MCMC sampling steps, the NFE of this setting is around 700, which is smaller than the 1000 NFE of DPS. All of the experiments are carried out on a single Nvidia A100 GPU. We report the running clock time of DPMC in Table 6 in Appendix C.

---

[1]https://github.com/LeviBorodenko/motionblur

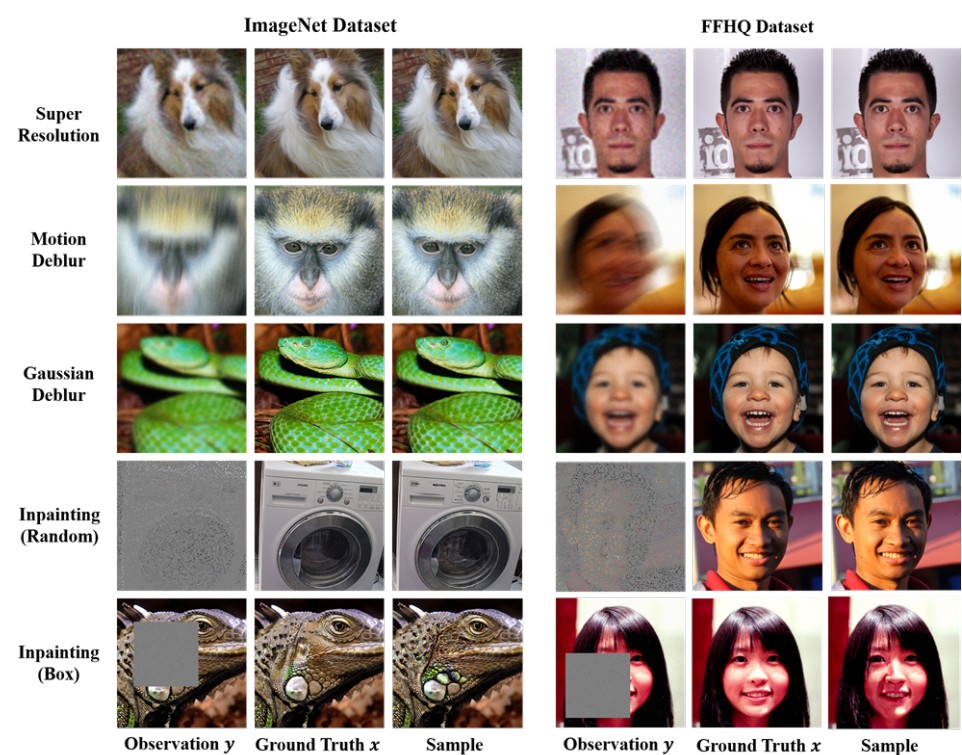

Figure 3: Qualitative results of DPMC on different linear inverse problem tasks.

**Baselines:** On linear inverse problems, we compare DPMC algorithm with the original DPS (Chung et al., 2023a), filter posterior sampling (FPS) with and without sequential Monte Carlo sampling (Dou & Song, 2023), ΠGDM Song et al. (2023b), DDNMWang et al. (2023), RED-Diff Mardani et al. (2023a), DiffPir Zhu et al. (2023a), CCDF Chung et al. (2022b), denoising diffusion restoration models (DDRM) (Kawar et al., 2022), manifold constrained gradients (MCG) (Chung et al., 2022a), plug-and-play alternating direction method of multiplier (PnP-ADMM) (Chan et al., 2017), Score-Based SDE (Song et al., 2021b; Choi et al., 2021) and alternating direction method of multiplier with total-variation (TV) sparsity regularized optimization (ADMM-TV). We get the results of FPS, DiffPir, CCDF, DDRM, MCG, PnP-ADMM, Score-Based SDE, ADMM-TV from the corresponding papers. We reproduce the results of ΠGDM and RED-Diff with the official implementation [2] of RED-Diff. And We reproduce the results of DDNM using its official implementation [3].

On phase retrieval, we compare DPMC with DPS (Chung et al., 2023a), oversampling smoothness (OSS) (Rodriguez et al., 2013), Hybrid input-output (HIO) (Fienup & Dainty, 1987) and error reduction (ER) (Fienup, 1982) algorithm.

## 4.2 EXPERIMENTAL RESULTS

**Linear Inverse Problems:** We carried out experiments on FFHQ and ImageNet datasets. We show qualitative samples of each task in Figure 3. More results can be found in Appendix . DPMC is able to generate valid samples given the noisy, degraded input. Note that, as discussed in 2.1, given the information loss in the image degradation process, exactly restoring the original $\mathbf{x}$ is ill-posed. Instead, an effective algorithm should be able to fill in meaningful content that agrees with the noisy observation. This is particularly true for tasks with large information loss, such as inpainting large areas in the image with random or box masks. We show an example in Figure 4, where we present samples generated with different random seeds using the same noisy observation $\mathbf{y}$ in the inpainting

---

[2]https://github.com/NVlabs/RED-diff
[3]https://github.com/wyhuai/DDNM

task with a box-shaped mask. DPMC is capable of generating various samples that agree with the observed part.

Qualitative results are shown in Table 1 and Table 2. Following Chung et al. (2023a); Dou & Song (2023), we report the LPIPS score (Zhang et al., 2018) and FID (Heusel et al., 2017) score. The LPIPS score measures the similarity of predicted samples with the ground truth at the single image level. As discussed in Zhang et al. (2018), unlike PSNR and SSIM, which capture shallow, low-level features and might fail to account for many nuances of human perception, LPIPS focuses more on structured information related to human perception. On the other hand, FID measures the distribution differences between generated samples and observations. We believe these two metrics are well-suited to reflect sample performance given the many-to-one mapping nature between $\mathbf{y}$ and $\mathbf{x}$.

We have observed that the results of RED-diff and ΠGDM are significantly influenced by whether noise is added to the degradation process. We observed that if noise is added, the optimized samples might exhibit noise-like artifacts and sometimes completely fail. Therefore, we include both the noisy version (same as our setting) and the noiseless version (denoted as $\sigma = 0$) here. Note that due to setting differences, the noiseless results are listed here for reference only.

As we can see, DPMC achieves similar or better results than DPS on both datasets across all tasks using less NFE. This is also evident in the qualitative comparisons in Figure 2, where DPS provides blurred samples and our DPMC fills in more vivid details. Compared to other strong baselines, such as FPS, $\pi$GDM, DDNM, DPMC achieves superior results on most tasks, especially in terms of FID. A full qualitative comparison of DPMC and these baselines can be found in figure 11. DPMC has the advantage in the sample details and will not provide samples containing noise-like artifacts in noisy settings ($\sigma > 0$). This demonstrates the effectiveness of introducing MCMC in the sampling process.

**Nonlinear Inverse Problem:** We conducted the phase retrieval experiment on the FFHQ dataset. For this, we utilized $T = 200$ intermediate distributions and $K = 6$ MCMC sampling steps at each intermediate distribution. The proposal-exploration step was still applied to the middle 60% of the sampling steps, corresponding to 920 NFE. Similar to DPS, we observed that the quality of final samples depend on the initialization. Therefore, we followed DPS by generating four different samples and selecting the best one. Our qualitative results are shown in Figure 5, and the quantitative results are reported in Table 3. Compared to DPS, our DPMC achieved better LPIPS and a similar FID score.

Table 1: Quantitative results of various linear inverse problems (with $\sigma = 0.05$) on FFHQ $256 \times 256$-1k validation set. **Bold** denotes the best result for each task and underline denotes the second best result. Results with "*" are reproduced by ourselves. We also list the noiseless version ($\sigma = 0$) of RED-Diff and ΠGDM on the bottom for reference.

| Methods | Super Resolution | | Inpainting (box) | | Gaussian Deblur | | Inpainting (random) | | Motion Deblur | |
|---|---|---|---|---|---|---|---|---|---|---|
| | FID | LPIPS | FID | LPIPS | FID | LPIPS | FID | LPIPS | FID | LPIPS |
| DPMC (Ours) | **21.93** | 0.212 | **19.59** | 0.160 | **21.34** | **0.210** | 21.26 | 0.205 | **20.73** | 0.213 |
| DDNM*(Wang et al., 2023) | 26.64 | 0.214 | 25.97 | 0.150 | 28.69 | 0.212 | 28.71 | **0.201** | - | - |
| RED-Diff* (Mardani et al., 2023a) | 89.13 | 0.435 | - | - | 37.35 | 0.255 | - | - | - | - |
| ΠGDM* (Song et al., 2023b) | 29.59 | 0.214 | - | - | 431.83 | 0.887 | - | - | - | - |
| CCDF (Chung et al., 2022b) | 60.90 | - | 49.77 | - | - | - | - | - | - | - |
| DiffPir (Zhu et al., 2023a) | - | 0.260 | - | 0.236 | - | - | - | - | - | 0.255 |
| FPS (Dou & Song, 2023) | 26.66 | 0.212 | 26.13 | **0.141** | 30.03 | 0.248 | 35.21 | 0.265 | 26.18 | 0.221 |
| FPS-SMC (Dou & Song, 2023) | 26.62 | **0.210** | 26.51 | 0.150 | 29.97 | 0.253 | 33.10 | 0.275 | 26.12 | 0.227 |
| DPS (Chung et al., 2023a) | 39.35 | 0.214 | 33.12 | 0.168 | 44.05 | 0.257 | **21.19** | 0.212 | 39.92 | 0.242 |
| DDRM (Kawar et al., 2022) | 62.15 | 0.294 | 42.93 | 0.204 | 74.92 | 0.332 | 69.71 | 0.587 | - | - |
| MCG (Chung et al., 2022a) | 87.64 | 0.520 | 40.11 | 0.309 | 101.2 | 0.340 | 29.26 | 0.286 | - | - |
| PnP-ADMM (Chan et al., 2017) | 66.52 | 0.353 | 151.9 | 0.406 | 90.42 | 0.441 | 123.6 | 0.692 | - | - |
| Score-SDE (Song et al., 2021b; Choi et al., 2021) | 96.72 | 0.563 | 60.06 | 0.331 | 109.0 | 0.403 | 76.54 | 0.612 | - | - |
| ADMM-TV | 110.6 | 0.428 | 68.94 | 0.322 | 186.7 | 0.507 | 181.5 | 0.463 | - | - |
| RED-Diff* ($\sigma = 0.0$) (Mardani et al., 2023a) | 39.68 | 0.185 | - | - | 30.54 | 0.161 | - | - | - | - |
| ΠGDM* ($\sigma = 0.0$) (Song et al., 2023b) | 39.61 | 0.207 | - | - | 34.52 | 0.140 | - | - | - | - |

## 4.3 ABLATION STUDY

We conducted ablation studies on critical parameters of DPMC using the Gaussian deblur task on the FFHQ dataset. Table 4a examines the impact of the number of MCMC sampling steps. Table 4b explores the influence of the number of intermediate distributions. Table 4c evaluates different weighting schedules by setting $\xi_t \propto f(\bar{\alpha}_t)$, where $f(\cdot)$ ranges from a constant function to $\bar{\alpha}_t^4$. For each ablation setting, we adjusted the task-related parameters $\eta$ and $\xi$ to optimize the current configuration. The results demonstrate that using more sampling steps or intermediate distributions

Table 2: Quantitative results of various linear inverse problems (with $\sigma = 0.05$) on ImageNet $256 \times 256$-1k validation set. **Bold** denotes the best result for each task and underline denotes the second best result. Results with "*" are reproduced by ourselves. We also list the noiseless version ($\sigma = 0$) of RED-Diff and ΠGDM on the bottom for reference.

| Methods | Super Resolution | | Inpainting (box) | | Gaussian Deblur | | Inpainting (random) | | Motion Deblur | |
|---|---|---|---|---|---|---|---|---|---|---|
| | FID | LPIPS | FID | LPIPS | FID | LPIPS | FID | LPIPS | FID | LPIPS |
| DPMC (Ours) | **31.74** | **0.307** | **30.55** | 0.221 | **33.62** | **0.318** | **30.25** | **0.292** | **30.88** | **0.303** |
| DDNM*(Wang et al., 2023) | 41.62 | 0.317 | 33.91 | 0.208 | 37.25 | 0.321 | 37.82 | 0.315 | - | - |
| RED-Diff* (Mardani et al., 2023a) | 82.62 | 0.471 | - | - | 39.11 | 0.319 | - | - | - | - |
| ΠGDM* (Song et al., 2023b) | 43.55 | 0.343 | - | - | 371.33 | 0.813 | - | - | - | - |
| DiffPir (Zhu et al., 2023a) | - | 0.371 | - | 0.355 | - | - | - | - | - | 0.366 |
| FPS (Dou & Song, 2023) | 47.32 | 0.329 | 33.19 | **0.204** | 54.41 | 0.396 | 42.68 | 0.325 | 52.22 | 0.370 |
| FPS-SMC (Dou & Song, 2023) | 47.30 | 0.316 | 33.24 | 0.212 | 54.21 | 0.403 | 42.77 | 0.328 | 52.16 | 0.365 |
| DPS (Chung et al., 2023a) | 50.66 | 0.337 | 38.82 | 0.262 | 62.72 | 0.444 | 35.87 | 0.303 | 56.08 | 0.389 |
| DDRM (Kawar et al., 2022) | 59.57 | 0.339 | 45.95 | 0.245 | 63.02 | 0.427 | 114.9 | 0.665 | - | - |
| MCG (Chung et al., 2022a) | 144.5 | 0.637 | 39.74 | 0.330 | 95.04 | 0.550 | 39.19 | 0.414 | - | - |
| PnP-ADMM (Chan et al., 2017) | 97.27 | 0.433 | 78.24 | 0.367 | 100.6 | 0.519 | 114.7 | 0.677 | - | - |
| Score-SDE (Song et al., 2021b; Choi et al., 2021) | 170.7 | 0.701 | 54.07 | 0.354 | 120.3 | 0.667 | 127.1 | 0.659 | - | - |
| ADMM-TV | 130.9 | 0.523 | 87.69 | 0.319 | 155.7 | 0.588 | 189.3 | 0.510 | - | - |
| RED-Diff* ($\sigma = 0$) (Mardani et al., 2023a) | 45.17 | 0.304 | - | - | 32.29 | 0.232 | - | - | - | - |
| ΠGDM* ($\sigma = 0$) (Song et al., 2023b) | 50.21 | 0.342 | - | - | 32.99 | 0.200 | - | - | - | - |

| **Ground Truth $x$** | **Observation $y$** | **Sample 1** | **Sample 2** | **Sample 3** |
|---|---|---|---|---|

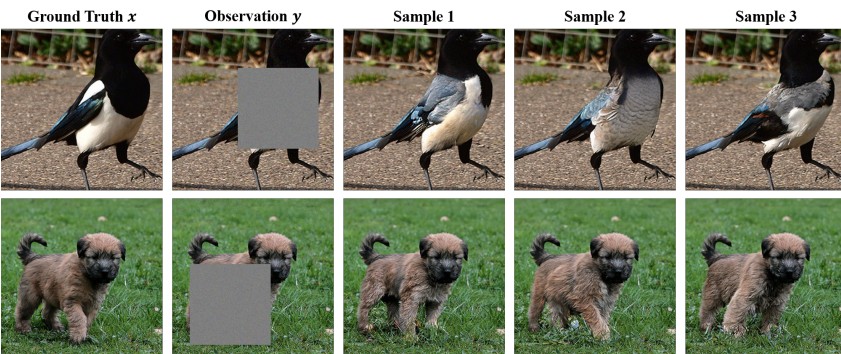

Figure 4: Inpainting results with box-shape mask using different random seed. The first column is the ground truth. The second column is the masked observation. The third to fifth columns are results get by our algorithm under different random seeds.

improves performance, with $T = 200$ and $K = 4$ being a good choice for balancing performance and sample efficiency. Additionally, setting $\xi_t = \xi \bar{\alpha}_t^3$ yields better results compared to constant scheduling or other alternatives.

Table 4: Ablation studies on number of MCMC sampling step $K$, number of intermediate distribution $T$ and $\epsilon_t$ schedule. In 4a, we keep the same number of intermediate distributions and change the number of MCMC sampling steps; in 4b, we keep the same MCMC sampling steps and change the number of intermediate distributions; in 4c, we keep the number of MCMC steps and the number of intermediate distributions and change different weighted schedule $\xi_t$.

(a) Keep $T = 200$, Change $K$

| | **K=1** | **K=2** | **K=4** | **K=6** |
|---|---|---|---|---|
| **LPIPS** | 0.227 | 0.214 | 0.210 | 0.209 |
| **FID** | 26.81 | 22.00 | 21.34 | 21.25 |

(b) Keep $K = 4$, Change $T$

| | **T=100** | **T=200** | **T=300** | **T=400** |
|---|---|---|---|---|
| **LPIPS** | 0.220 | 0.210 | 0.209 | 0.207 |
| **FID** | 21.89 | 21.34 | 21.29 | 21.35 |

(c) Keep $T = 200$, $K = 4$, Change $\xi_t$ schedule.

| | **Constant** | $\bar{\alpha}_t$ | $\bar{\alpha}_t^2$ | $\bar{\alpha}_t^3$ | $\bar{\alpha}_t^4$ |
|---|---|---|---|---|---|
| **LPIPS** | 0.232 | 0.219 | 0.213 | 0.210 | 0.209 |
| **FID** | 24.72 | 22.53 | 21.62 | 21.34 | 21.43 |

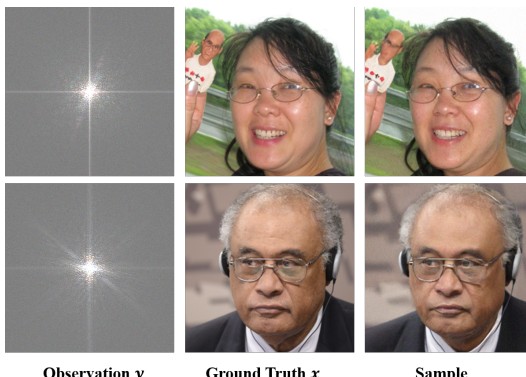

**Observation _y_**  **Ground Truth _x_**  **Sample**

Figure 5: Qualitative results of DPMC on FFHQ $256 \times 256$ phase retrieval task.

Table 3: Quantitative results of DPMC on FFHQ $256 \times 256$ phase retrieval task.

| Methods | FID | LPIPS |
|---|---|---|
| DPMC (Ours) | 55.64 | **0.372** |
| DPS (Chung et al., 2023a) | **55.61** | 0.399 |
| OSS (Rodriguez et al., 2013) | 137.7 | 0.635 |
| HIO (Fienup & Dainty, 1987) | 96.40 | 0.542 |
| ER (Fienup, 1982) | 214.1 | 0.738 |

## 5 CONCLUSION AND FUTURE WORK

In this study, we propose DPMC, an algorithm based on a specific formula of the posterior distribution and Annealed MCMC sampling to solve inverse problems. We demonstrate that DPMC results in superior sample quality compared to DPS across various inverse problems with fewer functional evaluations. Our study underscores the benefit of incorporating more sampling steps into each intermediate distribution to encourage exploration. Additionally, it is beneficial to trade in the number of intermediate distributions with the number of MCMC exploration steps. One potential limitation of our current work is the necessity to manually tune the weighting schedule and other hyper-parameters, whose optimal values can vary for different tasks. An interesting future direction is to explore scenarios where an explicit estimation $p(\mathbf{x})$ is provided by an EBM or another likelihood-based estimation technique, enabling the use of more advanced samplers such as Hamiltonian Monte Carlo (HMC) with adaptive step sizes. As a powerful algorithm that can process images, DPMC might have the potential to cause negative social consequences, including deepfakes, misinformation, and privacy breaches. We believe that more research and resources are needed to mitigate these risks.

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

## A  EXPERIMENTAL DETAILS

In this study, for linear inverse problem, we use $T = 200$ intermediate distributions and $K = 4$ MCMC sampling step at each intermediate distributions as our default setting. For the nonlinear inverse problem, phase retrieval, we use T = 200 intermediate distributions and K = 6 MCMC sampling step at each intermediate distributions. Our main task specific parameters are the Guidance weight $\xi$ and Langevin step size coefficient $\eta$. We show the parameter setting for each task in Table 5. All of our experiments are carried on a single Nvidia A100 GPU.

Table 5: Hyper-parameter settings for various linear inverse problems. $\xi$: Guidance Weight; $\eta$: Langevin Step Size Coefficient

(a) FFHQ dataset

| Settings | $\xi$ | $\eta$ |
|---|---|---|
| Super Resolution | 3.6e3 | 0.2 |
| Inpainting (box) | 4.2e3 | 0.5 |
| Gaussian Deblur | 6e3 | 0.3 |
| Inpainting (random) | 6e3 | 0.5 |
| Motion Deblur | 6.6e3 | 0.3 |
| Phase Retrieval | 1.5e3 | 3.0 |

(b) ImageNet dataset

| Settings | $\xi$ | $\eta$ |
|---|---|---|
| Super Resolution | 3.3e3 | 0.4 |
| Inpainting (box) | 5.1e3 | 0.4 |
| Gaussian Deblur | 4.5e3 | 0.3 |
| Inpainting (random) | 5.7e3 | 0.5 |
| Motion Deblur | 6e3 | 0.5 |

## B  THEORETICAL ANALYSIS

In this section, we provide some theoretical analysis on the DPMC model proposed by us. First, we start with some existing results on the KL-convergence of the Langevin MCMC algorithm. Denote $p^*$ as the target distribution over $\mathbb{R}^d$, and $s^*(\mathbf{x}) := \nabla \log p^*(\mathbf{x})$ as its score function. The Langevin MCMC algorithm with step size $\eta$ is given by:

$$\mathbf{X}_0 \sim p_0, \quad \mathbf{X}_{i+1} = \mathbf{X}_i + \eta s^*(\mathbf{X}_i) + \sqrt{2\eta} \cdot \xi_i$$

where $\xi_i \sim \mathcal{N}(0, \boldsymbol{I}_d)$. Before we state the convergence rate, we propose the strong convexity and Lipschitz continuity assumption that the target distribution $p^*$ needs to satisfy.

**Assumption B.1** (Strong Convexity and Lipschitz Continuity of Potential Function). Let $U(\mathbf{X}) = -\log p^*(\mathbf{X})$ be the potential function. It has $m$-strong convexity and its gradient has $L$-Lipschitz continuous, i.e.:

$$m\boldsymbol{I}_d \preceq \nabla^2 U(\mathbf{X}) \preceq L\boldsymbol{I}_d \quad \text{for } \forall \mathbf{X} \in \mathbb{R}^d.$$

Under this assumption, Cheng & Bartlett (2018) propose the following total variation convergence result of the Langevin MCMC algorithm.

**Lemma B.2** (TV-Convergence of Langevin MCMC Algorithm). *After choosing the step size* $\eta = \frac{m\varepsilon^2}{32Ld^2}$, *and the number of iterations*

$$K = \frac{32L^2 d \log(D_{\mathrm{TV}}(p_0 \| p^*)/\varepsilon)}{m^2 \varepsilon^2},$$

*the last iterate distribution* $p_K := \mathrm{Law}(\mathbf{X}_K)$ *holds that:*

$$D_{\mathrm{TV}}(p_K \| p_0) \leq \varepsilon.$$

Notice that in the original Theorem 1 of Cheng & Bartlett (2018), the authors set $p_0 = \mathcal{N}(0, \frac{1}{m}\boldsymbol{I}_d)$. However, the proof does not actually rely on this choice. Besides, we use the corollary of TV-distance convergence instead of the original KL-divergence result since TV distance holds triangular inequality, which benefits our analysis.

Next, we study the distribution estimation error and provide a convergence guarantee for DPMC proposed by us. Our main result requires the following assumptions on the conditional distribution $p(\mathbf{x}_t \mid \mathbf{y})$ as well as the unconditional score estimation error.

**Assumption B.3** (Unconditional Score Estimation Error). For all $k = 1, 2, \ldots, N$, it holds that:

$$\mathbb{E}_{\mathbf{x} \sim p_{kh}} \|\hat{s}_\theta(\mathbf{x}, kh) - \nabla \log p_{kh}(\mathbf{x})\|^2 \leq \varepsilon_{\text{score}}^2$$

where $\hat{s}_\theta(\mathbf{x}, t)$ is the pretrained score estimator we plug into our algorithm, and $h = T/N$ is the time step.

**Assumption B.4** (Conditional Score Approximation). For all $k = 1, 2, \ldots, N$, we have an upper bound for the TV-distance between the true conditional distribution $p_{kh}(\mathbf{x}_{kh} \mid \mathbf{y})$ and our DPS-type approximation $\bar{p}_{kh}(\mathbf{x}_{kh} \mid \mathbf{y})$ as follows:

$$D_{\text{TV}}\left(\bar{p}_{kh}(\mathbf{x}_{kh} \mid \mathbf{y}) \,\|\, p_{kh}(\mathbf{x}_{kh} \mid \mathbf{y})\right) \leq \varepsilon_{\text{cond}}.$$

**Assumption B.5** (Lipschitz Continuity, Strongly Convexity and Bounded Moment of Conditional Score). For all $t$, the conditional score $\nabla \log p_t(\mathbf{x}_t \mid \mathbf{y})$ is $L$-Lipschitz continuous, and the potential function $\log p_t(\mathbf{x}_t \mid \mathbf{y})$ is $m$-strongly convex, which enables us to obtain exponential convergence of Langevin MCMC algorithm. We also assume that

$$\mathbb{E}_{\mathbf{x}_t \sim p_t(\cdot | \mathbf{y})} \|\nabla \log p_t(\mathbf{y} \mid \mathbf{x}_t)\|^2 \leq U_{\text{cond}}^2.$$

For $t = 0$, $p_0(\cdot \mid \mathbf{y})$ has bounded second-order moment, i.e.

$$m_2^2 := \mathbb{E}_{\mathbf{x}_0 \sim p_0(\cdot | \mathbf{y})} \|\mathbf{x}_0\|^2 < \infty.$$

It also leads to the conclusion that $p_0(\mathbf{x}_0 \mid \mathbf{y})$ has a bounded KL-divergence from the standard Gaussian distribution, i.e.

$$D_{\text{KL}}\left(p_0(\mathbf{x}_0 \mid y) \| \mathcal{N}(0, \boldsymbol{I}_d)\right) \leq \text{poly}(d).$$

**Assumption B.6** (Initial Conditional Gap). The TV-distance in Assumption B.4 is upper bounded by $\varepsilon_0$ when $t = 0$, i.e.

$$D_{TV}(\bar{p}_0(x_0|y), p(x_0|y)) \leqslant \varepsilon_0.$$

Unlike the Jensen gap $\varepsilon_{\text{cond}}$ which is expected to be large because DPS-type approximation $\bar{p}_{kh}(\mathbf{x}_{kh} \mid \mathbf{y})$ is usually not a good estimate of the true conditional distribution, $\varepsilon_0$ is only dependent on score estimation error, which can be sufficiently small after enough training.

Now, we state our main theorem as follows:

**Theorem B.7.** *Under Assumptions B.3-B.6, once our time step $h < 1/L \wedge 1$, we can guarantee that our last-iterate conditional distribution $q_0(\cdot \mid \mathbf{y})$ derived from DPMC is $(\varepsilon + \varepsilon_{\text{cond}})$-close from ground truth with respect to TV-distance:*

$$D_{\text{TV}}(q_0(\mathbf{x}_0 \mid \mathbf{y}), p_0(\mathbf{x}_0 \mid \mathbf{y})) \leq \varepsilon + \varepsilon_0$$

*after choosing the step size $\eta$ and the number of inner loops (denoted by $K$) of the Langevin MCMC algorithm at all time steps as follows:*

$$\eta = \frac{m\varepsilon^2}{32Ld^2}, \quad K = \frac{32L^2 d \cdot \log((\sqrt{\text{poly}(d) \cdot \exp(-T)} + \varepsilon_{\text{cond}})/\varepsilon \ \vee \ \varepsilon_{\text{inter}}/\varepsilon)}{m^2 \varepsilon^2}.$$

*Here,*

$$\varepsilon_{\text{inter}} := \varepsilon + \varepsilon_{\text{cond}} + C\sqrt{h} \cdot (L\sqrt{dh} + Lm_2 h) + C\sqrt{h}(\varepsilon_{\text{score}} + U_{\text{cond}}),$$

*and $C$ is a universal constant.*

*Proof.* Denote $q_t(\cdot)$ as the probability measure derived by the backward process of diffusion model. For the initial step of backward process, we have $q_T = \mathcal{N}(0, \boldsymbol{I}_d)$, whose distance from $p_T$ shows the convergence of forward process. According to Chen et al. (2022), the variance-preserving framework leads to exponential convergence of forward process, i.e.

$$D_{\text{KL}}\left(p_T(\mathbf{x}_T \mid y) \| \mathcal{N}(0, \boldsymbol{I}_d)\right) \leq D_{\text{KL}}\left(p_0(\mathbf{x}_0 \mid y) \| \mathcal{N}(0, \boldsymbol{I}_d)\right) \cdot \exp(-T) \leq \text{poly}(d) \cdot \exp(-T).$$

By using Pinsker's Inequality and Assumption B.4, we have:

$$D_{\text{TV}}\left(\bar{p}_T(\mathbf{x}_T \mid y) \| \mathcal{N}(0, \boldsymbol{I}_d)\right) \leq \sqrt{\text{poly}(d) \cdot \exp(-T)} + \varepsilon_{\text{cond}}.$$

Starting from $\tilde{\mathbf{x}}_T \sim q_T = \mathcal{N}(0, \boldsymbol{I}_d)$, we apply Langevin MCMC algorithm with regard to the score function $\nabla \log p_T(\mathbf{x}_T \mid \mathbf{y})$. By using Lemma B.2, we can make $D_{\mathrm{TV}}(q_T(\mathbf{x}_T \mid \mathbf{y}), \bar{p}_T(\mathbf{x}_T \mid \mathbf{y})) \leq \varepsilon$ by using step size $\eta = \frac{m\varepsilon^2}{32Ld^2}$, and the number of iterations

$$K = \frac{32L^2 d \log(D_{\mathrm{TV}}(q_T \| \bar{p}_T(\mathbf{x}_T \mid \mathbf{y}))/\varepsilon)}{m^2 \varepsilon^2}.$$

Next, we apply unconditional backward step as well as Langevin MCMC inner loops to make sure $D_{\mathrm{TV}}(q_{kh}(\mathbf{x}_{kh} \mid \mathbf{y}), \bar{p}_{kh}(\mathbf{x}_{kh} \mid \mathbf{y})) \leq \varepsilon$ holds for $\forall k \in [N]$, including the last iterate $k = 0$. We prove it by the method of induction. Assume it holds that $D_{\mathrm{TV}}\left(q_{(k+1)h}(\mathbf{x}_{(k+1)h} \mid \mathbf{y}) \| \bar{p}_{(k+1)h}(\mathbf{x}_{(k+1)h} \mid \mathbf{y})\right) \leq \varepsilon$, then we immediately have

$$D_{\mathrm{TV}}\left(q_{(k+1)h}(\mathbf{x}_{(k+1)h} \mid \mathbf{y}) \| p_{(k+1)h}(\mathbf{x}_{(k+1)h} \mid \mathbf{y})\right) \leq \varepsilon + \varepsilon_{\mathrm{cond}}$$

by using Assumption B.4 as well as the triangular inequality of TV distance. After applying a backward diffusion step with unconditional score estimator $\hat{s}_\theta(\cdot, t)$ plugged in and obtain $\bar{\mathbf{x}}_{kh}$. During this backward step, the score estimation error is actually

$$\mathbb{E}_{\mathbf{x}_{kh} \sim p_{kh}(\mathbf{x}_{kh}|\mathbf{y})} \| \hat{s}_\theta(\mathbf{x}_{kh}) - \nabla \log p_{kh}(\mathbf{x}_{kh} \mid \mathbf{y}) \|^2$$
$$\leq \mathbb{E}_{\mathbf{x}_{kh}} \| \hat{s}_\theta(\mathbf{x}_{kh}) - \nabla \log p_{kh}(\mathbf{x}_{kh}) - \nabla \log p_{kh}(\mathbf{y} \mid \mathbf{x}_{kh}) \|^2$$
$$\leq 2\mathbb{E}_{\mathbf{x}_{kh}} \| \hat{s}_\theta(\mathbf{x}_{kh}) - \nabla \log p_{kh}(\mathbf{x}_{kh}) \|^2 + 2\mathbb{E}_{\mathbf{x}_{kh}} \| \nabla \log p_{kh}(\mathbf{y} \mid \mathbf{x}_{kh}) \|^2$$
$$\leq 2(\varepsilon_{\mathrm{score}}^2 + U_{\mathrm{cond}}^2) \leq 2(\varepsilon_{\mathrm{score}} + U_{\mathrm{cond}})^2.$$

Therefore, we substitute the score estimation error $\varepsilon_{\mathrm{score}}$ in Theorem 2 of Chen et al. (2022) with $\varepsilon_{\mathrm{score}} + U_{\mathrm{cond}}$. We use the results in Chen et al. (2022) as well as Girsanov Theorem, and conclude that:

$$D_{\mathrm{TV}}\left(\mathrm{Law}(\bar{\mathbf{x}}_{kh}) \| p_{kh}(\mathbf{x}_{kh} \mid \mathbf{y})\right) - D_{\mathrm{TV}}\left(q_{(k+1)h}(\mathbf{x}_{(k+1)h} \mid \mathbf{y}) \| \bar{p}_{(k+1)h}(\mathbf{x}_{(k+1)h} \mid \mathbf{y})\right)$$
$$\leq \underbrace{C\sqrt{h} \cdot (L\sqrt{dh} + Lm_2 h)}_{\text{discretization error}} + \underbrace{C\sqrt{h}(\varepsilon_{\mathrm{score}} + U_{\mathrm{cond}})}_{\text{score estimation error}}$$

where $C$ is a universal constant, which leads to

$$D_{\mathrm{TV}}\left(\mathrm{Law}(\bar{\mathbf{x}}_{kh}) \| p_{kh}(\mathbf{x}_{kh} \mid \mathbf{y})\right) \leq \varepsilon + \varepsilon_{\mathrm{cond}} + C\sqrt{h} \cdot (L\sqrt{dh} + Lm_2 h) + C\sqrt{h}(\varepsilon_{\mathrm{score}} + U_{\mathrm{cond}}) := \varepsilon_{\mathrm{inter}}.$$

As the initial step of Langevin MCMC inner loops at time $t = kh$, we apply the exponential convergence (Lemma B.2) and show that we can make $D_{\mathrm{TV}}(q_{kh}(\mathbf{x}_{kh} \mid \mathbf{y}), \bar{p}_{kh}(\mathbf{x}_{kh} \mid \mathbf{y})) \leq \varepsilon$ and complete the induction by letting the step size $\eta = \frac{m\varepsilon^2}{32Ld^2}$ and the number of iterations

$$K = \frac{32L^2 d \cdot \log(\varepsilon_{\mathrm{inter}}/\varepsilon)}{m^2 \varepsilon^2}.$$

To sum up, we can guarantee that $D_{\mathrm{TV}}(q_0(\mathbf{x}_0 \mid \mathbf{y}), \bar{p}_0(\mathbf{x}_0 \mid \mathbf{y})) \leq \varepsilon$ by choosing the step size $\eta$ and the number of inner loop $K$ of Langevin MCMC algorithm as follows:

$$\eta = \frac{m\varepsilon^2}{32Ld^2}, \quad K = \frac{32L^2 d \cdot \log((\sqrt{\mathrm{poly}(d) \cdot \exp(-T)} + \varepsilon_{\mathrm{cond}})/\varepsilon \ \vee \ \varepsilon_{\mathrm{inter}}/\varepsilon)}{m^2 \varepsilon^2}$$

where

$$\varepsilon_{\mathrm{inter}} := \varepsilon + \varepsilon_{\mathrm{cond}} + C\sqrt{h} \cdot (L\sqrt{dh} + Lm_2 h) + C\sqrt{h}(\varepsilon_{\mathrm{score}} + U_{\mathrm{cond}}).$$

We have $D_{\mathrm{TV}}(\bar{p}_1(\mathbf{x}_1 \mid \mathbf{y}) \| p_1(\mathbf{x}_1 \mid \mathbf{y})) \leq \varepsilon_{\mathrm{cond}}$. For the final step, we do not have conditional Jensen gap. According to Assumption B.6, it finally comes to our conclusion as

$$D_{\mathrm{TV}}(\bar{p}_0(\mathbf{x}_0 \mid \mathbf{y}) \| p_0(\mathbf{x}_0 \mid \mathbf{y})) \leq \varepsilon + \varepsilon_0.$$

$\square$

## C MORE EXPERIMENTAL RESULTS

### C.1 SAMPLING TIME

In Table 6, we report the time for generating one sample with DPMC default setting, which sets $T = 200$, $K = 4$ and applies MCMC sampling steps in the middle 60% intermediate distributions. (See section 4.1 for more details.) The time is tested on the FFHQ dataset using a single Nvidia A100 GPU.

Table 6: Running Times of Different Methods for Generating one Sample on FFHQ-1k Validation Dataset

| Model | Running Time (Seconds) |
|---|---|
| DPMC(ours) | 54.63 |
| FPS Dou & Song (2023) | 33.07 |
| FPS-SMC (with $M = 5$) Dou & Song (2023) | 57.88 |
| DPS Chung et al. (2023a) | 70.42 |
| Score-SDE Song et al. (2022) | 32.93 |
| DDRM Kawar et al. (2022) | 2.034 |
| MCG Chung et al. (2022a) | 73.16 |
| PnP-ADMM Chan et al. (2017) | 3.595 |
| ΠGDM Song et al. (2023b) | 33.18 |

Table 7: PSNR and SSIM results of various linear inverse problems on FFHQ $256 \times 256$-1k validation set. **Bold** denotes the best result for each task and underline denotes the second best result. Results with "*" are reproduced by ourselves.

| Methods | Super Resolution | | Inpainting (box) | | Gaussian Deblur | | Inpainting (random) | | Motion Deblur | |
|---|---|---|---|---|---|---|---|---|---|---|
| | PSNR | SSIM | PSNR | SSIM | PSNR | SSIM | PSNR | SSIM | PSNR | SSIM |
| DPMC (Ours) | 27.52 | 0.772 | 23.55 | 0.828 | 27.29 | 0.758 | 26.40 | 0.776 | **27.55** | 0.771 |
| DDNM Wang et al. (2023) | 28.60 | 0.766 | 24.15 | 0.829 | 27.18 | 0.750 | 26.88 | 0.762 | - | - |
| RED-diff Mardani et al. (2023b) | 27.07 | 0.686 | - | - | **30.34** | 0.799 | - | - | - | - |
| ΠGDM Song et al. (2023c) | **28.64** | 0.820 | - | - | 6.12 | 0.011 | - | - | - | - |
| FPS Dou & Song (2023) | 27.48 | 0.807 | 24.17 | 0.865 | 26.45 | 0.773 | 26.79 | 0.820 | 26.70 | 0.828 |
| FPS-SMC Dou & Song (2023) | 28.10 | 0.807 | **24.70** | 0.862 | 26.54 | 0.773 | **27.33** | 0.819 | 27.39 | 0.826 |
| DPS Chung et al. (2023a) | 25.67 | 0.852 | 22.47 | **0.873** | 24.25 | 0.811 | 25.23 | **0.851** | 24.92 | **0.859** |
| DDRM Kawar et al. (2022) | 25.36 | 0.835 | 22.24 | 0.869 | 23.36 | 0.767 | 9.19 | 0.319 | - | - |
| MCG Chung et al. (2022a) | 20.05 | 0.559 | 19.97 | 0.703 | 6.72 | 0.051 | 21.57 | 0.751 | - | - |
| PnP-ADMM Chan et al. (2017) | 26.55 | **0.865** | 11.65 | 0.642 | 24.93 | **0.812** | 8.41 | 0.325 | - | - |
| Score-SDE Song et al. (2021b); Choi et al. (2021) | 17.62 | 0.617 | 18.51 | 0.678 | 7.12 | 0.109 | 13.52 | 0.437 | - | - |
| ADMM-TV | 23.86 | 0.803 | 17.81 | 0.814 | 22.37 | 0.801 | 22.03 | 0.784 | - | - |
| RED-diff* (No noise $\sigma = 0$) | 30.75 | 0.875 | - | - | 33.43 | 0.910 | - | - | - | - |
| ΠGDM* (No noise $\sigma = 0$) | 29.36 | 0.842 | - | - | 36.77 | 0.940 | - | - | - | - |

## C.2 More Samples for Each Inverse Problems

In Figure 6, 8, 7, 9, 10, we show more samples for each inverse problem. DPMC works well across different tasks and datasets.

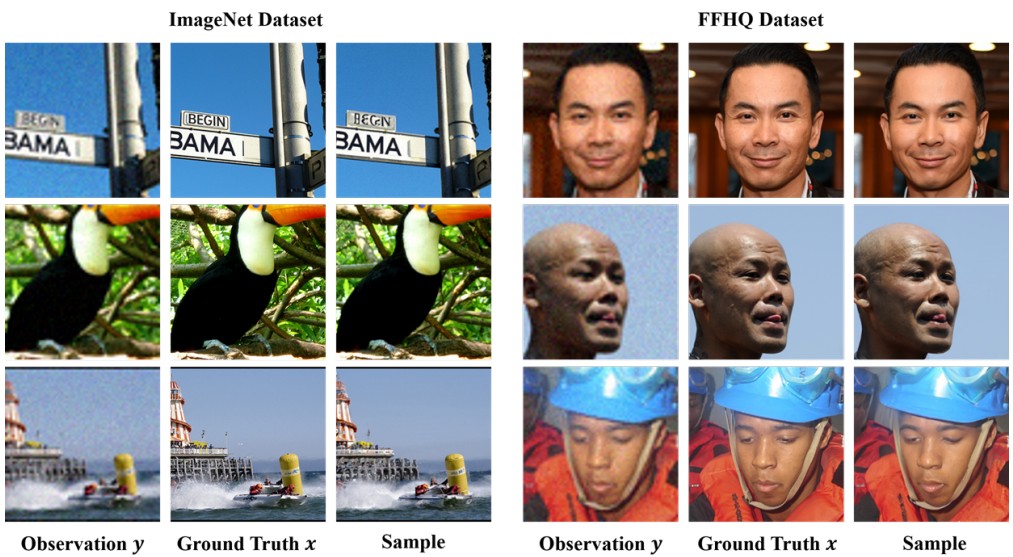

Figure 6: More Results for Super Resolution Task

Table 8: PSNR and SSIM results of various linear inverse problems on ImageNet $256 \times 256$-1k validation set. **Bold** denotes the best result for each task and underline denotes the second best result. Results with "*" are reproduced by ourselves.

| Methods | Super Resolution | | Inpainting (box) | | Gaussian Deblur | | Inpainting (random) | | Motion Deblur | |
|---|---|---|---|---|---|---|---|---|---|---|
| | PSNR | SSIM | PSNR | SSIM | PSNR | SSIM | PSNR | SSIM | PSNR | SSIM |
| DPMC (Ours) | 23.30 | 0.594 | 19.38 | 0.692 | 23.05 | 0.578 | 21.56 | 0.586 | **23.47** | 0.597 |
| DDNM Wang et al. (2023) | 24.29 | 0.589 | 21.72 | 0.707 | 24.07 | 0.585 | 22.98 | 0.579 | - | - |
| RED-diff Mardani et al. (2023b) | 23.25 | 0.555 | - | - | **25.72** | 0.659 | - | - | - | - |
| ΠGDM Song et al. (2023c) | 24.30 | 0.662 | - | - | 6.08 | 0.019 | - | - | - | - |
| FPS Dou & Song (2023) | 24.32 | 0.724 | 20.16 | 0.752 | 23.58 | 0.581 | 23.39 | 0.688 | 22.71 | 0.598 |
| FPS-SMC Dou & Song (2023) | **24.78** | 0.731 | **22.03** | 0.748 | 23.81 | 0.599 | **24.12** | 0.685 | 23.27 | 0.614 |
| DPS Chung et al. (2023a) | 23.87 | 0.781 | 18.90 | 0.794 | 21.97 | **0.706** | 22.20 | **0.739** | 20.55 | **0.634** |
| DDRM Kawar et al. (2022) | 24.96 | **0.790** | 18.66 | **0.814** | 22.73 | 0.705 | 14.29 | 0.403 | - | - |
| MCG Chung et al. (2022a) | 13.39 | 0.227 | 17.36 | 0.633 | 16.32 | 0.441 | 19.03 | 0.546 | - | - |
| PnP-ADMM Chan et al. (2017) | 23.75 | 0.761 | 12.70 | 0.657 | 21.81 | 0.669 | 8.39 | 0.300 | - | - |
| Score-SDE Song et al. (2021b); Choi et al. (2021) | 12.25 | 0.256 | 16.48 | 0.612 | 15.97 | 0.436 | 18.62 | 0.517 | - | - |
| ADMM-TV | 22.17 | 0.679 | 17.96 | 0.785 | 19.99 | 0.634 | 20.96 | 0.676 | - | - |
| RED-diff* (No noise $\sigma = 0$) | 25.10 | 0.701 | - | - | 27.94 | 0.796 | - | - | - | - |
| ΠGDM* (No noise $\sigma = 0$) | 24.62 | 0.675 | - | - | 31.86 | 0.876 | - | - | - | - |

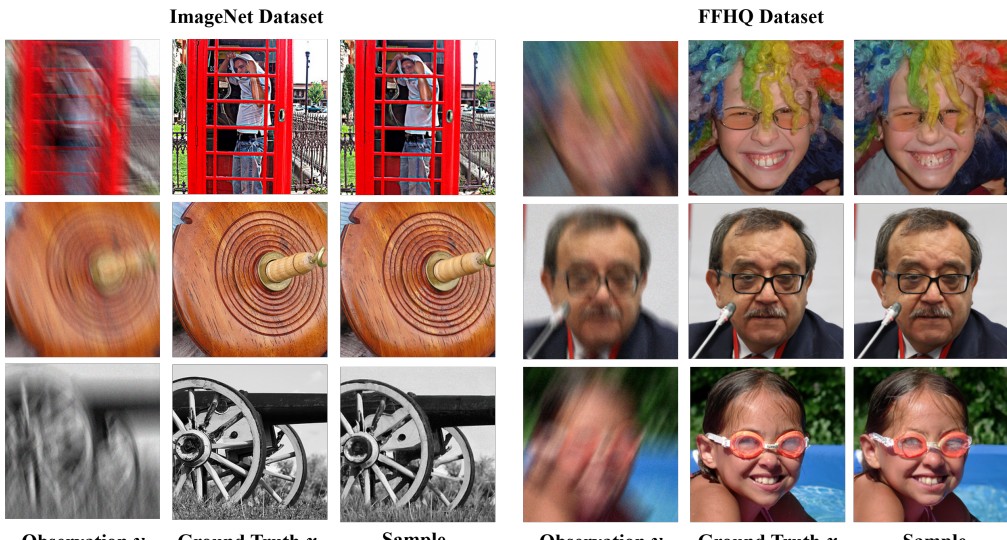

**ImageNet Dataset**        **FFHQ Dataset**

**Observation $y$**   **Ground Truth $x$**   **Sample**    **Observation $y$**   **Ground Truth $x$**   **Sample**

Figure 7: More Results for Motion Deblur Task

## C.3 PSNR AND SSIM

In Table 7 and Table 8, we report the PSNR and SSIM scores for a reference. As discussed in section 4.2, we chose to primarily compare models using FID and LPIPS in our paper for two main reasons. First, as discussed by Zhang et al. (2018), PSNR and SSIM capture shallow, low-level features and might fail to account for many nuances of human perception. Second, the inverse problem is an ill-posed problem. Due to information loss in the degradation process, there can be multiple possible solutions, and the model does not have to 100% faithfully reproduce all the details. As shown in the quantitative comparison of Figure 2 and Figure 11, our DPMC can fill in vivid details to the optimized samples. It is possible that these filled-in details are reasonable but not entirely the same as the input, resulting in lower scores on PSNR or SSIM. High level features like LPIPS and FID might work better in judging the quality of the generated samples.

ImageNet Dataset
FFHQ Dataset

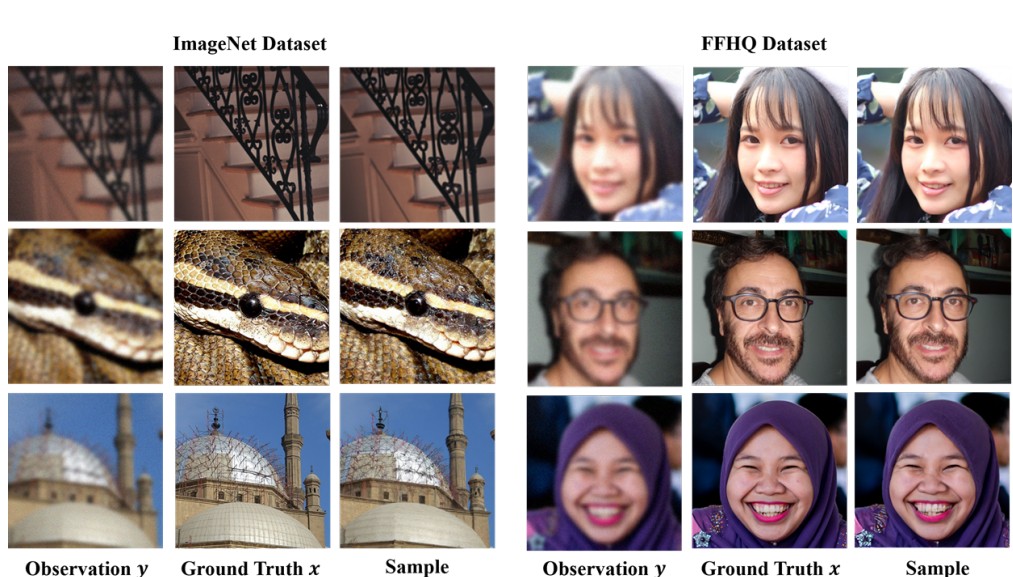

Observation $y$  Ground Truth $x$  Sample  Observation $y$  Ground Truth $x$  Sample

Figure 8: More Results for Gaussian Deblur Task

ImageNet Dataset
FFHQ Dataset

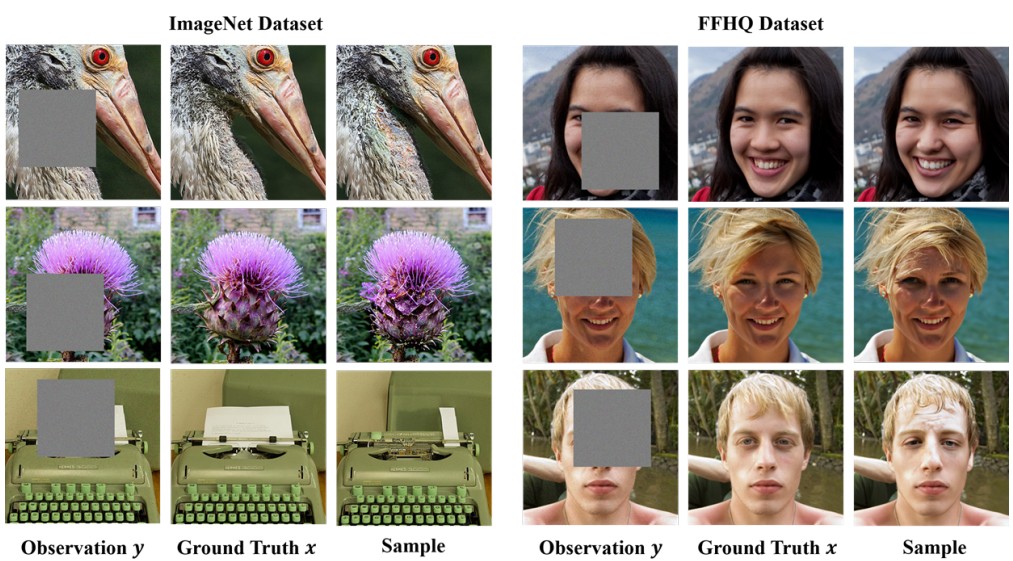

Observation $y$  Ground Truth $x$  Sample  Observation $y$  Ground Truth $x$  Sample

Figure 9: More Results for Inpainting (Box) Task

**ImageNet Dataset**  **FFHQ Dataset**

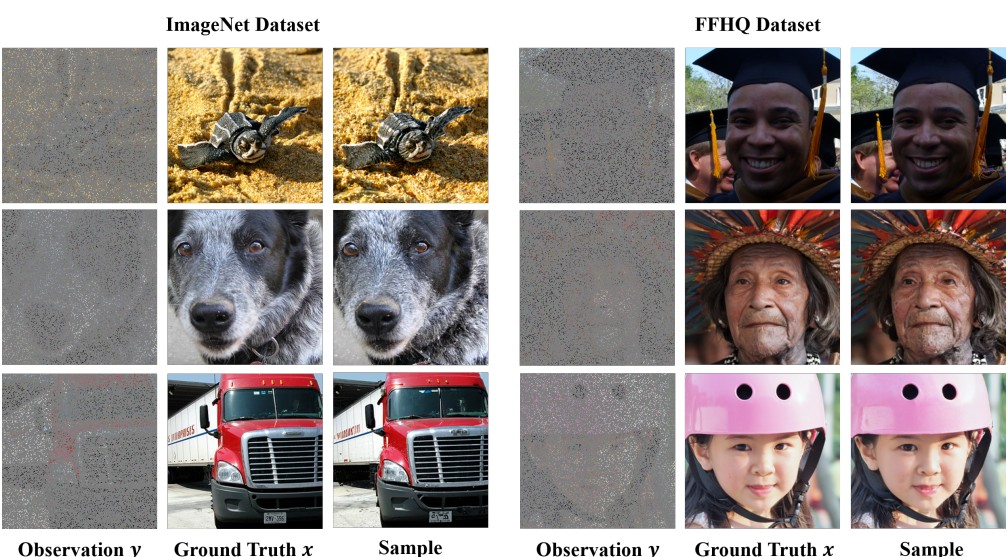

Observation $y$    Ground Truth $x$    Sample      Observation $y$    Ground Truth $x$    Sample

Figure 10: More Results for Inpainting (Random) Task

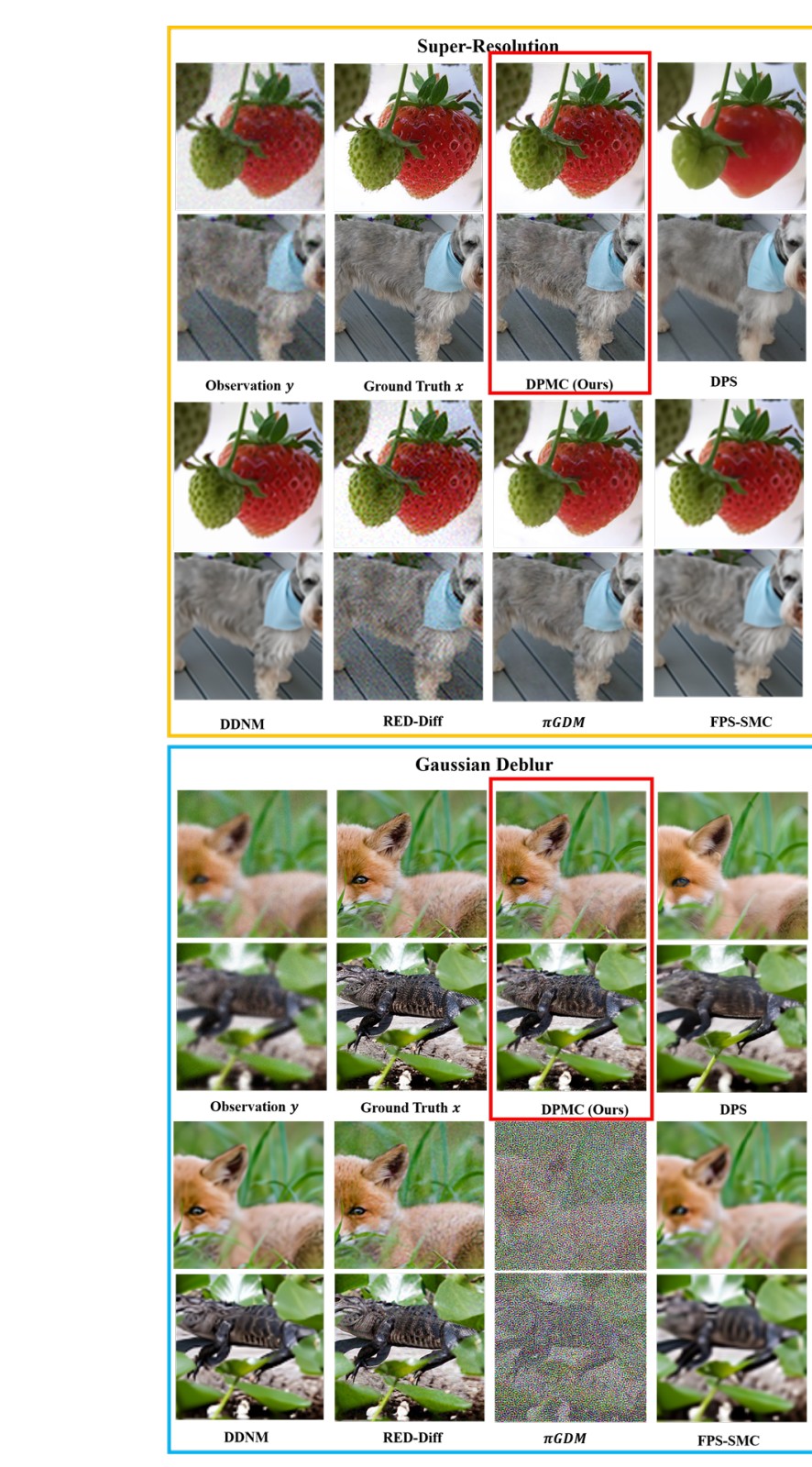

Figure 11: Qualitative comparison of different methods on ImageNet $256 \times 256$-1k dataset. We highlight the results of DPMC with red rectangles.

