# OpenReview forum: "Think Twice Before You Act: Improving Inverse Problem Solving With MCMC"
_ICLR.cc/2025/Conference — ICLR 2025 Conference Withdrawn Submission_

### Official Review · Reviewer_Y3XZ · 2024-10-28

**Soundness:** 3
**Presentation:** 3
**Contribution:** 2
**Rating:** 6
**Confidence:** 4

**Summary:**

The authors present a method based on annealed MCMC for solving inverse problems using diffusion models. The motivation for this work stems from the fact that existing methods like DPS [Chung et al.] rely on overly simplistic approximations of the diffusion posterior $p(x_0|x_t)$ throughout the diffusion sampling procedure when solving inverse problems. In this work, alternatively, the authors propose to sample from intermediate distributions by introducing additional MCMC correction steps. Empirical results are presented on different linear and non-linear inverse problems on the FFHQ and ImageNet datasets at 256x256 resolution.

**Strengths:**

**Originality**: While the proposed idea of using annealed MCMC has been utilized for unconditional diffusion sampling in several prior works [Song et al.], its application to solving inverse problems combined with existing posterior approximation methods is novel.

**Quality and Clarity**: The paper is written clearly, and the choice of baselines is adequate. The empirical results in Tables 1 and 2 are comprehensive regarding the selection of datasets and inverse problems. The ablation studies take into account the impact of different design choices in the method.

**Significance**: Though the computational overhead of introducing MCMC corrections as an inner loop in iterative diffusion sampling seems much, the method has significance in terms of yielding better quality generations at similar sampling budgets as DPS, which could be useful in scenarios when a large sampling budget is available.

**Weaknesses:**

**Requested Changes/Questions**

1. **On the choice of the intermediate distribution in Eq. 9**: The authors use the posterior approximation from DPS to define the intermediate distributions as stated in Eq. 9. In principle, the approximation from Pi-GDM can also be used to formulate the same. What is the rationale behind sticking with the approximation in DPS since Pi-GDM can perform better with fewer NFE? It would be great if the authors could clarify this choice in the main text.

2. **On the choice of the proposal**: The authors propose to initialize MCMC sampling by generating a sample from a single denoising step from the diffusion model without guidance. Intuitively, it looks like initializing the MCMC chain with denoising + guidance could yield better samples since the latter might be closer to the target intermediate distribution. Is this choice motivated by reducing the computational overhead incurred by initializing using denoising + guidance, as the guidance term requires computing a Jacobian-vector product? It would be great if the authors could clarify more on this or share some experimental evidence that justifies the choice of their current proposal over the one mentioned here.

3. In Eq. 11, the authors suggest to use the  l2-norm over the squared l2-norm. How critical is this choice? Moreover, it is unclear if the same choice is also used for the vanilla DPS baseline for a fair comparison since the authors specify using a squared l2 norm in Eq. 7. It would be nice if the authors could present a small ablation for one inverse problem on the dataset of their choice.

4. **Computational Overhead**: By introducing an inner MCMC loop, the computational overhead of this method increases significantly, especially since each MCMC correction step requires a Jacobian-vector product through the score network. The authors propose to counteract this by reducing the number of intermediate distributions to around 200, which results in a total number of NFE to around 700. Though this is smaller than DPS, it is quite a lot compared to some recent work [1,2] in solving inverse problems in diffusion. While I won't stress on comparing DPMC to these methods, it would be nice to discuss this aspect of inverse problems in the paper (maybe in the main text or the appendix) as a possible extension of this work. Moreover, can the authors extend the results in Table 4b to say T=25 or 50 to assess the impact on performance at lower sampling budgets? Lastly, I would expect K to decrease as the signal-to-noise ratio increases in diffusion sampling since lower MCMC corrections should be required as sampling progresses. Did the authors experiment with a schedule on K, or is the choice of a fixed K in the paper primarily motivated by simplicity constraints? Perhaps the last section of the paper can benefit from this discussion.

[1] Fast Samplers for Inverse Problems in Iterative Refinement Models, Pandey et al.

[2] Accelerating Diffusion Models for Inverse Problems through Shortcut Sampling, Liu et al.

**Minor Comments/Corrections**

1. In some places, like Line 354, method names and citations are hard to decipher. I think readability can be improved by using a different color for citation links.

2. Tables 1 and 2 can benefit from indicating that lower is better for lpips and fid.

3. Line 369, the link to the appendix is broken.

4. There are a lot of missing entries in Table 1. Can the authors clarify in the caption if the results were unavailable for these cases or if the methods were unstable?

**Questions:**

See the above section.

---

### Official Review · Reviewer_Zv1M · 2024-10-30

**Soundness:** 2
**Presentation:** 3
**Contribution:** 2
**Rating:** 5
**Confidence:** 4

**Summary:**

Diffusion Posterior Sampling (DPS) is a versatile algorithm for solving inverse problems with pre-trained diffusion models. However, DPS suffers from bias due to the Dirac delta approximation of the noise-conditional posterior $p(x_0|x_t) \approx \delta(x_0 - E[x_0|x_t]),$ which can lead to large errors even for a simple mixture of Gaussians. To address this, the authors propose Diffusion Posterior MCMC (DPMC), which reduces bias by defining a series of intermediate distributions inspired by the approximated conditional distributions in DPS.
In its simplest form, DPMC can be viewed as performing multiple DPS steps with added noise. Through annealed MCMC sampling, the intermediate latents closely follow the carefully constructed intermediate distributions before moving to the next noise level. Extensive experiments on FFHQ 256x256 and ImageNet 256x256 demonstrate DPMC’s effectiveness on five inverse problems: super-resolution, box inpainting, random inpainting, Gaussian deblurring, and motion deblurring.

**Strengths:**

1. This paper aims to reduce the bias in diffusion posterior sampling by incorporating MCMC sampling into the intermediate steps of the reverse diffusion process.

2. The experimental results are promising across a range of inverse tasks, including super-resolution (4x), random inpainting, motion deblurring, Gaussian deblurring, and box inpainting.

**Weaknesses:**

1. In the Line 6 of Algorithm 1, the update follows Eq 10. The second term in Eq 10 requires $\nabla \log p_t(x_t)$ and the gradient of the measurement error. How is the score function $\nabla \log p_t(x_t)$ computed using the pre-trained neural network? Since the NN expects the state and time as inputs, and time changes after every update, how is the time step (or equivalently the noise level) chosen while computing the score?

2. The proposed DPMC sampler is a stochastic equivalent of RB-Modulation (Algorithm 1). Substituting the gradient of the proposal distribution in Eq. (10):
$$x_{t-1}^{k+1} = x_{t-1}^k + \eta_{t-1} \nabla_{x_{t-1}^k} \log p_{t-1}(x_{t-1}^k) -  \rho  \eta_{t-1} \nabla_{x_{t-1}^k} || y - A \hat{x}(x_{t-1}^k) ||^2 + \sqrt{2 \eta_{t-1}} w.$$
Without the noise term $w$, DPMC is equivalent to RB-Modulation (Algorithm 1, https://arxiv.org/pdf/2405.17401 ). While $w$ helps in exploration, it slows down the inference process due to large mixing time. On the other hand, RB-Modulation aims for the maximum likelihood estimate and converges much faster. Both the algorithms make certain approximations to reduce inference time but the core idea remains the same. The authors are encouraged to clarify the distinction between these two algorithms and highlight the algorithmic novelty of DPMC over prior work RB-Modulation.

3. Contributions 2 and 3 could be combined, as they convey a similar message.

4. The first equality in Eq. (2) is incorrect. It should be properly factorized to accurately represent the probabilities.

5. Inconsistent notation used in Line 136 for $p_t(x_t)$. With Bayes rule, it should just be a marginal of $p$.

6. Typo: Missing log in Line 269.

7. The primary motivation of this paper is that $p(x_0|x_t)$ may be multi-modal; hence, replacing  $x_0 \sim p(x_0|x_t)$ with its conditional expectation $\hat{x}_0$ at higher noise levels could produce misleading results. However, the authors still use $\hat{x}_0$ in the first 30\% of the reverse process, which corresponds to high noise levels – contradicting the main motivation.

8. What happens if these initial 30\% steps mislead the generation process? Does the proposed sampling mechanism rectify this issue in subsequent steps? If not, the technical benefits of these MCMC steps must be clearly explained, rather than better storytelling via MCMC sampling.

**Questions:**

Please see the weaknesses above.

---

### Official Review · Reviewer_pK1v · 2024-11-03

**Soundness:** 3
**Presentation:** 2
**Contribution:** 2
**Rating:** 5
**Confidence:** 4

**Summary:**

The paper addresses an interesting topic: solving inverse problems using diffusion models. The authors argue that the performance of the popular DPS algorithm is limited by inaccuracies in the posterior approximation, particularly at high noise levels. To address this, they propose an MCMC algorithm to sample more accurately from the posterior distribution.

**Strengths:**

1. The theoretical bound presented in the appendix is insightful.
2. The paper is well-structured and easy to follow (altough there are some problems with writing as I explained in the weakness part).
3. The experimental results demonstrate the superiority of the proposed method over state-of-the-art alternatives, both quantitatively and qualitatively.
4. The experiments are thorough and consider relevant datasets and settings.

**Weaknesses:**

1. **Marginal Contribution**: The contribution of the paper appears limited, as it primarily applies the established MCMC algorithm within diffusion models for solving inverse problems. Specifically, the novelty seems to lie in the use of Equation (10) atop standard diffusion models, where Equation (10) serves as the MCMC update step.

2. **Method Clarity**: The authors should dedicate more space to clearly explaining their method in Section 3, especially detailing the deployment of MCMC in posterior sampling. Currently, the description on page 5 is divided into two parts (Proposal Stage and Exploration Stage), which makes the algorithm challenging to follow. Mathematical expressions should be used alongside explanations to clarify each step. Specific areas for improvement include:

   a) **Proposal Stage Description**: The description of the proposal stage is vague. For instance, the phrase "..we first denoise them to \( t-1 \) following..." is unclear. Does \( t-1 \) refer to \( x_{t-1} \)?

   b) **Standard Diffusion Step**: When the authors mention the "standard diffusion step," are they referring to using a pre-trained diffusion model to estimate \( x_t \)?

   c) **Understanding of the Sampling Process**: My understanding is that the method first uses an unconditional pre-trained diffusion model to estimate \( x_{t-1} \) and then modifies \( x_{t-1} \) to sample from the posterior. If this is correct, the writing should be adjusted to make this process explicitly clear in the paper.

3. **Terminology on Line 248**: On line 248, the authors state that "..The samples \( \tilde{x}_{t-1} \) might not fully adhere to the target distribution \( \tilde{p}_{t-1}(x_{t-1}|y) \)." Given that samples drawn from the unconditional distribution would indeed differ significantly from those conditioned on \( y \), the use of "might not fully adhere" is imprecise. The terminology should be refined here to convey a more scientifically rigorous statement.

**Questions:**

Please see my comments above.

---

### Official Review · Reviewer_7SHa · 2024-11-10

**Soundness:** 3
**Presentation:** 2
**Contribution:** 2
**Rating:** 3
**Confidence:** 5

**Summary:**

The authors propose a new algorithm for solving inverse problems with pre-trained diffusion models. The proposed algorithm is a variation of Diffusion Posterior Sampling (DPS) where instead of taking one gradient step to match the measurements, the authors propose running Langevin dynamics to sample from the approximate conditional distribution. The authors demonstrate the empirical performance of their algorithm for linear and non-linear inverse problems.

**Strengths:**

1) The empirical results are quite strong. The authors demonstrate that their method outperforms natural baselines across different inverse problems such as mask inpainting, box inpainting, super-resolution, deblurring, and phase retrieval.

2) The algorithmic modification is simple, which might allow for wide adoption from the community.

3) The research direction of developing algorithms for solving inverse problems with diffusion models is relevant and interesting.

**Weaknesses:**

1) I believe that the presentation of the paper could be improved. There are several examples. In Line 43, I believe that measure should be changed to measurement. In Lines 46-47, I believe that the authors want to say that the posterior is intractable (it is always defined, but it is intractable to sample from it or write down an explicit formula for its density). I also believe that it is a little confusing to what former and later refer to in lines 158-159. The theoretical result is not clearly stated, there is no mention in the main paper of the assumptions and on what epsilon depends to and how.

2) Apart from not being clearly stated, the theoretical result is very weak (it has very strong assumptions such as strong convexity) and I do not understand the value it offers to the paper. My guess is that under all these strong assumptions on the data distribution, DPS would also converge to the true distribution (given properly tuned learning rates and a small enough discretization step).

3) I believe that the authors overcomplicate the presentation of their algorithm and as far as I understand what's happening is an inner loop of Langevin Dynamics that is getting interleaved between the denoising steps. Langevin Dynamics is the simplest algorithm in the MCMC family. Having an inner loop that enforces measurement consistency in the diffusion sampling has been proposed previously in the literature. For example, the papers Beyond First Order Tweedie and Resample also seem to solve an inner optimization problem between each denoising step.

4) The fact that the authors only use the proposed method for a subset of the steps makes the evaluation harder.

5) I believe that the paper would benefit from some latent diffusion experiments since there are powerful latent diffusion models available that can/should be leveraged to solve inverse problems.

**Questions:**

See also weaknesses above.  Explicit questions:

1) It is not clear whether this method performs better for high corruption or for low corruption. Could the authors provide a plot where the horizontal axis is the corruption level and the vertical axis the performance and show how their method compares with DPS across these different regimes?

2) Could the authors provide comparisons of what happens if both DPS and the proposed method have big enough computational budget? If NFEs are not a concern, does the method still outperform DPS?

3) Could the authors clarify whether the theoretical results would also hold for DPS under similar assumptions and learning rate tuning?

4) Could the authors compare with other methods that perform similar inner loop optimizations (such as Resample and Beyond First Order Tweedie)?

---

### Note · Authors · 2024-11-19

**Comment:**

We sincerely appreciate the reviewers' valuable comments and suggestions. While we have confidence in our proposed method, unforeseen circumstances during the rebuttal period have made it challenging for our main authors to complete the requested experiments and analyses within the limited timeframe. Therefore, we have decided to withdraw our submission at this time. We will carefully consider the feedback and incorporate it to improve our work for future submissions.

**Withdrawal Confirmation:**

I have read and agree with the venue's withdrawal policy on behalf of myself and my co-authors.